# The Effects of an 8-Week Cognitive–Motor Training Program on Proprioception and Postural Control Under Single and Dual Task in Older Adults: A Randomized Clinical Trial

**DOI:** 10.3390/healthcare12222297

**Published:** 2024-11-17

**Authors:** Ainhoa Nieto-Guisado, Monica Solana-Tramunt, Cristina Cabrejas, Jose Morales

**Affiliations:** Department of Sport Sciences, FPCEE Blanquerna, Ramon Llull University, 08022 Barcelona, Spain; ainhoang@blanquerna.url.edu (A.N.-G.); monicast2@blanquerna.url.edu (M.S.-T.); cristinacm19@blanquerna.url.edu (C.C.)

**Keywords:** motor control, balance, active aging, training with music, double task training

## Abstract

The aim of this study was to assess the effects of an 8-week cognitive-motor training program on postural control and knee proprioception under single and dual task conditions. **Design**: Randomized clinical trial. **Methods**: The present study was registered with the ID number NCT04786132. A total of 20 healthy and physically active older adults (73.25 ± 5.98 years) volunteered to participate and were randomly assigned into an experimental and a control group (EG and CG). Postural control was measured with the Romberg test, with open (RBOE) and closed eyes (RBCE) and under unipodal dominant side (RUDL) conditions. Proprioception was assessed by measuring participants’ ability to reposition their dominant knee at 45°. Finally, performance of the cognitive task was measured through a subscale of the Barcelona Test called “categorical evocation in associations”. The EG and the CG completed 8-week training programs with two sessions, 30 min per week, of postural control and proprioception exercises. The EG additionally included music in each session. **Results:** The analysis using a mixed ANOVA model revealed no significant group × time interaction effects (*p* < 0.05) for any of the assessments. However, a significant main effect for the time factor was observed, with both the control and experimental groups showing improved outcomes in the post-intervention measurements. Specifically, significant results were found for RBOE (F (4,15) = 11.87, *p* < 0.001, η^2^*p* = 0.76), RBCE (F (4,15) = 11.62, *p* < 0.001, η^2^*p* = 0.75), and proprioception (F (1,18) = 11.53, *p* < 0.003, η^2^*p* = 0.39). **Conclusions:** The 8-week training program had a positive impact on the post-intervention results for motor control and proprioception, but not on the results of the cognitive task. There were no significant differences between the groups that carried out sessions with or without music.

## 1. Introduction

The increased longevity of the current population has sparked researchers’ interest in studying the potential of different modalities of physical activity-based programs to delay the physical and cognitive decline associated with aging, with the aim of improving the quality of life and overall well-being of the older adult population [1]. Experts predict a potential increase in the proportion of the population over 60 years of age in the near future [2]. Therefore, it is necessary to examine the influence of different kinds of exercise programs on the functioning of the central nervous system (CSN), as well as on the perceptual–motor qualities that depend on it. Indeed, it is believed that certain kinds of physical training can improve people’s level of motor control and cognitive functioning and, in turn, help them lead more effective everyday lives [3].

It is in this daily life in which we carry out our daily activities where the central nervous system (CNS) has to process information from different stimuli that can be both motor and cognitive, sometimes separately and other times simultaneously [4,5]. This work of receiving information, processing it, and developing an appropriate response is called a task [4]; the frequency with which this information is processed interferes with the functioning of the brain that has to process it. A simple task is defined as one in which a single task to be performed by the subject is prioritized, either motor or cognitive [4]. However, when two tasks that can be carried out independently and measured separately, and may even have different objectives, must be performed simultaneously, it is called double task or “dual task” [5]. Double tasking is common in daily life since there are many occasions when a motor task is preceded by a cognitive task or vice versa, and in addition, we normally carry out activities that involve the simultaneous performance of a cognitive function and a motor task [5].

If we talk about a motor skill or task, we understand one that involves a response controlled directly or indirectly by cognitive processes consciously or unconsciously and occurring in the form of muscle contraction, for example, postural control or a response to proprioceptive afferents [6]. To contextualize the different terms that will be discussed throughout the text, proprioception is defined as the functional state of the joints and muscles that allows us to be aware of the movement or position of a part of the body or to respond unconsciously to the involuntary modification of our joint position in order to maintain postural control, muscle tone, or muscle coordination [7]. Meanwhile, postural control is defined as the ability to maintain the center of body mass within the base of support [8]. Postural control is achieved using information received from the vestibular apparatus and the inner ear, as well as other visual, pressure, tactile, and proprioceptive information [9], thus causing proprioceptive work to positively interfere with their own performance. Elsewhere, the concept of cognitive ability has been defined as the skills and the processes within the brain that allow us to receive, process, and elaborate upon information to carry out a task [10]. This includes attention, understanding, elaboration, and memorization of said information [11]. These cognitive processes are crucial for adapting our postural responses to varying environments and challenges, ensuring that we can maintain stability and balance effectively and consequently interfering with our daily activities.

During the aging process, there is a decrease in the above-mentioned motor and cognitive abilities, either due to the appearance of diseases such as Alzheimer’s or to factors associated with a person’s lifestyle and lifelong physical activity habits [12]. This is why performing more everyday social activities can lead to the improvement of motor and cognitive skills, given that performing single or double tasks has an effect on these skills [4]. Recent studies confirm that a higher educational and occupational level has a direct relationship with a high degree of Cognitive Reserve (CR) [12], which is the capacity that the brain develops to more efficiently tolerate the effects of a greater number of neuropathologies (such as those associated with dementia) before the symptoms manifest clinically [13]. CR develops naturally through the brain’s activity during our daily experiences and is shaped by various factors like education, job, living location, and sports participation, among others [14,15]. This is why a higher CR is linked to better education, which also suggests a greater level of both physical and mental activity. People with more education often enjoy additional benefits in adulthood, such as better access to resources and chances to engage in both physical and mental activities, which enhances motor–cognitive skills further [12,16].

Given the link between physical and cognitive tasks, performing physical exercise simultaneously with cognitive exercise improves motor and cognitive parameters, making it possible for the older adult population to delay the onset of diseases such as Alzheimer’s or dementia or to slow the development of their symptoms [17,18,19]. Furthermore, work on joint postural control and proprioception is one of the main objectives of researchers studying the older adult population because work in this capacity has been demonstrated to reduce the risk of falls in older people [20,21], thus reducing the risk of injury to the lower limbs [22], in addition to improving balance and gait [23].

The literature on motor–cognitive training indicates that a dual training program should last between 8 and 12 weeks and feature two to three sessions a week, of 15 to 45 min in duration, in order for it to have a positive effect [24]. Such training initiatives are based on imbalance and walking exercises, combined with cognitive exercises such as counting, word memorization or the Stroop test, or music and rhythm; they have been shown to improve performance on motor and cognitive control tests [18,25]. Music is employed by numerous scholars to engage individuals afflicted with cognitive or motor disorders, including Alzheimer’s and Parkinson’s disease, among others, owing to its inherently motivational characteristics, which foster enhanced adherence among this demographic to dual training regimens, simultaneously augmenting self-esteem during critical phases of the illness [26]. Moreover, in the elderly population, music exerts a positive influence due to its multifaceted nature; the act of listening to lyrics or musical compositions stimulates diverse cerebral regions, thereby addressing elements such as memory and concentration, while concurrently necessitating attention to rhythm, which is instrumental in maintaining balance, spatial awareness, and coordination with peers, thus impacting fall prevention, as sustaining a consistent rhythm aids in the rhythmic modulation of ambulation [25]. This enhancement exhibits a positive correlation with optimal motor and cognitive functioning, postponing the decline of these systems and facilitating improvements in fundamental executive functions that are crucial for daily living. Furthermore, engaging in musical activities, such as singing or playing instruments, fosters social interaction and emotional expression, which are vital for mental health and well-being, ultimately contributing to a more enriched quality of life for older adults [23,27,28].

Currently, many researchers are focusing on older adults and how training can improve their quality of life. Studying exercise among healthy seniors is important because it can greatly enhance their quality of life and help prevent chronic diseases. Regular exercise boosts muscle strength, stability, and heart health, which lowers the risk of falls and fractures and helps maintain independence. Additionally, it positively affects mental and brain health, reducing issues like depression and cognitive decline. These studies are essential for tailoring exercise programs that promote active aging [29,30,31]. A recent systematic review concludes that dual task exercise is more effective than single task exercise at reducing the risk of falls in healthy adults [32]. Prior studies have documented typical training programs to prevent falls in older adults, most of them focused on people with pathologies such as Parkinson’s, Alzheimer’s, dementia, or knee joint problems that cause them to fall more easily [24,32].

Despite this, there exists a paucity of information and a limited number of studies concerning the integration of dual motor–cognitive tasks within training sessions aimed at enhancing the quality of life of healthy and physically active older adults, which do not exclusively concentrate on falls or cognitive deficits. Consequently, we postulate that the incorporation of the dual motor–cognitive task in training sessions will yield enhancements in performance on the postural control and proprioception assessments (both abilities are related to each other, due to the influence that proprioceptive work has on improving postural control), conducted with a simultaneous cognitive task after the training period. For that reason, the aim of the present investigation was to assess the impact of an 8-week cognitive–motor training regimen, which was supplemented by a concurrent cognitive task, on the postural control and knee proprioception of healthy and physically active older adults, and our final hypothesis is that the experimental group will improve postural control and proprioception tests performed with a dual task after 8 weeks of dual motor–cognitive training conducted with sung music, compared to the control group that performs training without music, exclusively motor.

## 2. Materials and Methods

### 2.1. Design

A randomized clinical trial was used to determine the effects of the inclusion of a dual motor–cognitive task in postural control and proprioception training.

### 2.2. Participants

The study included 20 healthy and physically active older adults (17 male and 3 female) [according to the results of the International Physical Activity Questionnaire (IPAQ) 3912.52 ± 1738.57; Metabolic Equivalents (METS)] aged 73.25 ± 5.98 years and with 56.8 ± 16.78 CR. They were recruited in a Barcelona sport center during October of 2019 and volunteered to participate in the study. Participants were randomly allocated to either the experimental group (EG, *n* = 10) or the control group (CG, *n* = 10) using the online tool Research Randomizer (randomizer.org, accessed on 25 September 2019). Exclusion criteria included (i) suffering from a neurological condition such as Alzheimer’s or Parkinson’s disease, (ii) an inability to perform one or more of the parts of the test, (iii) being on medication that altered the normal functioning of the nervous system, and/or (iv) having recently had lower limb joint replacement surgery (in the 12 months prior to the tests).

The G*Power software (version 3.1, Bonn FRG, University of Bonn, Department of Psychology) was used to calculate the post hoc power analysis to know the statistical power of a mixed model ANOVA test to calculate within–between interactions and simple effects of the comparison of 2 groups (experimental and control) and 2 measurements (pre and post) of a total sample of 20 participants, effect size = 0.4 and α = 0.05, obtaining a power (1 − β) = 0.92.

The study was carried out in accordance with the ethical standards established in the latest version of the Declaration of Helsinki, and it was approved by the ethics committee of the Ramon Llull University, ID number 1718006D. The trial was registered as a current randomized controlled trial on clinicaltrials.gov with the ID number NCT04786132. All participants in the study were informed of the procedure and signed informed consent documents.

### 2.3. Proprioception

To evaluate proprioception, the Absolute Error of joint repositioning at 45° (AEr45°) of the knee of the dominant leg was calculated with a valid and reliable mobile application called Goniometer Pro, installed on a smartphone Galaxy J7 (Samsung, Seoul, South Korea). Using this application, previously validated [33], the mobile device can be used as a digital goniometer (2.9, FiveFufFive Co, Bloomfield, NJ, USA), providing instantaneous, accurate, and repeatable readings of the range of movement (ROM), to measure the difference between the requested and developed knee angle [34].

Firstly, orthopedic boots (Figure 1) were placed on each of the subjects’ legs in order to cancel the adaptation made by the toes and ankle and focus on the exclusive use of the knee. Next, the researchers placed the smartphone in alignment with the longitudinal of the femur, with the base of the device aligned to the interline of the femorotibial joint.

Although in previous works ski boots were used to limit the movement of the toes and ankle [35], in this study the ski boot was replaced with an orthopedic one, due to the ease of putting on and removing the boot, in addition to being able to match all subjects equally in terms of sizes or support, since the boots had adjustable straps that adapted to any foot size, thus restricting plantar flexion and dorsiflexion.

Once the subjects were equipped with the instruments, they closed their eyes and the researchers placed them in a 45° knee flexion position and instructed them to maintain that position for six seconds. Subsequently, the subjects were asked to recover the initial standing position and, when they were ready, to flex their knees again until they reached the position in which they believed they had been previously positioned by the researcher with their eyes closed. The repositioning angles on the dominant knee were noted by repeating the measurement 3 times and recording the average of the results. The absolute value of the difference between the requested angle (45°) and the one realized was recorded, to determine the joint repositioning capacity of the participants, a variable through which conscious proprioception is quantified [34].

### 2.4. Postural Control

Postural control was assessed with a Kistler force platform (Kistler Instruments AG, Winterthur, Switzerland) connected to a laptop computer running Kistler MARS 3.0 software. To evaluate bipodal and unipodal postural control, subjects were asked to stand on the platform and perform the Romberg test in 3 different conditions: Romberg bipodal open eyes (RBOE); Romberg bipodal closed eyes (RBCE); and Romberg unipodal dominant leg (RUDL). Prior to carrying out the test, participants received guidelines. They were told to stay upright with their gaze fixed on a point and their arms extended laterally close to the trunk. They were also asked to place their feet with at a width similar to that of their shoulders for the bipodal test. In the case of the unipodal postural control tests, the subjects were asked to perform a 45° knee flexion. As in the proprioceptive test, the subjects wore orthopedic boots on each of their legs to use the same conditions for the proprioceptive and postural control tests. The duration of each test was 30 s, and three repetitions were performed for each condition, with rest between repetitions of 30 s as well, to avoid muscle fatigue and give the subject time to recover physically and mentally.

The postural control variables that were obtained from the Center of Pressure (CoP) signals acquired by the force platform were the total area and the average velocity in the antero-posterior (MV_AP_) and medio-lateral (MV_ML_) directions. While the total area is an indicator of the performance of the postural control task, the MV_AP_ and MV_ML_ reveal the neuromuscular activity used to maintain postural control [36]. In all the variables studied, lower values are associated with better performance on the task.

The measurements of variables related to proprioception and postural control were carried out in a room free of disturbances and distractions. The order of carrying out the tests was randomized for each subject, and between tests the participant had a rest time of 3 min, to give the subjects time to get off the platform, have the next test explained, and begin.

### 2.5. Cognitive Control

To carry out the dual task performance, subjects performed a cognitive task while their postural control and proprioception were evaluated [37]. The cognitive task came from the subscale called “categorical evocation in associations”, which is part of the battery of subscales of the Revised Barcelona Test [38]. The task consisted of mentioning the maximum number of words possible for 30 s belonging to the same semantic field. The researchers determined the semantic fields so that they were known by all subjects regardless of age. The semantic fields chosen were the following: ***1. Animals; 2. Fruits; 3. Cities; 4. Parts of the body; 5. Clothing; 6. Meals; 7. Countries; 8. Colors; 9. Women’s names; 10. Men’s names*.** The score obtained depended on the number of words remembered, scoring one point for each of them. No points were added for repeated words or synonyms [38]. To quantify the total number of words that the subjects said, they were recorded, and once the tests were finished, the total number of words mentioned in each repetition was counted.

Each motor test was performed three times, so for each of the repetitions the subjects were asked about different semantic fields so that prior learning or short-term memory did not influence the motor results.

### 2.6. Procedure

A few days before carrying out the measurement protocol, the researchers visited the institutions from which the participants had been recruited, to collect data by anonymous forms sent by email to the participants. Prior to sending the email, we checked that older adults were familiar with the internet, and we gave them verbal instructions for filling out the form. This included information on the participants’ physical characteristics, along with data to determine whether the individuals fulfilled the inclusion/exclusion criteria; no medical history was collected. In accordance with [39], the participants’ lower limb dominance was established through a self-report on their performance on bilateral movement tasks involving the legs. To ensure that the order of the tests did not influence the results, the instruments were administered in a randomized order to each of the participants. In addition, the subjects answered a validated CR scale questionnaire which consisted of 24 questions related to their lifestyle to provide us with information about their lives [40]. Finally, each participant completed the short version IPAQ [41] to provide us with information about their self-reported degree of physical activity.

Once the tests were completed, the members of the sample were randomly assigned to the experimental EG and CG groups. After 8 weeks of intervention, the subjects were measured again on the postural control and proprioception tests with a cognitive task simultaneously (Figure 2).

### 2.7. Intervention Protocol

Once the pre-intervention measurements had been completed, an 8-week training protocol was carried out. The subjects, as previously mentioned, were randomly assigned into two groups, a control group and an experimental group.

Each participant of both the EG and CG conducted two sessions per week on different days to avoid contamination between the groups. Sessions lasted thirty minutes and consisted of a five-minute warm-up, a twenty-minute main section, and a five-minute cool-down period. During the main part of each session, the instructors worked to ensure that the exercises were technically executed in the correct manner in order to avoid injuries and to help promote the improvement of the participants’ postural control and proprioception.

The main part began with a series of general mobility exercises. Next, a simple choreography was taught. The steps were divided into five measures of thirty-two beats, each of them divided into four parts of eight movements. The cool-down period consisted of breathing exercises and basic stretching.

The difference between the training protocols of the EG and the CG was in the inclusion in the EG sessions of the singing of unknown music. Meanwhile, the CG performed the same exercises without musical accompaniment. The EG was asked to make the mental effort required to memorize and learn the songs that were playing in the first sessions and then to try to sing them while they were completing their motor exercises. Only two exercise choreographies were performed with the same music, one for the first four weeks and the other for the remaining weeks, in order to allow participants to become familiar with the lyrics. Meanwhile, the control group performed the motor exercises without music with the instructions of the person carrying out the session.

Four different sessions were designed (Appendix A). Each of these sessions was repeated for two full weeks during the 8 weeks, so the participants were able to learn to perform the various exercise techniques well to carry them out correctly. Sessions were designed to ensure that exercises became progressively more difficult, beginning with individual balance and proprioception exercises in the first session, then moving on to activities in pairs in the second, trios in the third, and groups of four in the fourth and until the end of the protocol. The proposed exercises included simple and complex walks, lateral movements, forward and backward movements, simple balances with both legs, and more complex ones performed along with classmates and with knee flexions of no more than 45° so that they would be related to the proprioception tests. For example, when performing a squat, imbalance activities were carried out with pairs or trios to compromise the stability of the participants, or participants were asked to perform dances such as the twist or the conga. This was in addition to working on balance and promoting socialization and fun for the participants.

Once the intervention was completed, the subjects performed the measurements again to help determine the effects of the intervention.

### 2.8. Statistical Analysis

Statistical analysis was performed using SPSS Version 21 software (SPSS Inc., Chicago, IL, USA). Prior to this analysis, the Kolmogorov–Smirnov test was applied to analyze the distribution of the variables. All variables showed a normal distribution, and, in order to verify the proposed hypotheses, a mixed model analysis of variance (ANOVA) was applied to evaluate the effects of an intra-subject factor (time: pre and post) and an inter-subject factor (group: control and experimental) under each of the execution conditions of postural control (open eyes (OE), closed eyes (CE), and unipodal leg (UDL)) and proprioception and cognitive tasks.

The descriptive data on the variables are presented as mean ± SD. Follow-up of multivariate contrasts was performed using univariate contrasts. When statistically significant main or interaction effects were found, pairwise comparisons were performed using the Bonferroni correction. This correction was applied to all results. A value of *p* < 0.05 was accepted as the level of significance for all statistical analyses.

## 3. Results

The multivariate contrasts in the RBOE condition showed a significant main effect of the time factor, indicating that both groups (control and experimental) recorded better results in the post-intervention measurements (F_4,15_ = 11.87, *p* < 0.001, η^2^*p* = 0.76). However, no significant group x time interaction effect was found; a t-student test was applied to verify that there were no differences between the groups in the initial phase, meaning that the results of the intervention program did not vary as a function of the group. The univariate analysis reported that the time effect was significant in all postural control variables in this condition but was not significant in the results of the cognitive task. The pairwise comparisons can be seen in Figure 3.

The multivariate analyses under the RBCE condition showed a significant main effect of the time factor, with better results registered in the post-intervention measurements for both groups (control and experimental) (F_4,15_ = 11.62, *p* < 0.001, η^2^*p* = 0.75). Nonetheless, as in the previous condition, no significant group x time interaction effect was found, so the intervention program had no differing effects as a function of the group. The univariate analysis revealed that the effect of time was significant in all postural control variables in this condition but was not significant when it came to the results of the cognitive task. The differences between pairs can be seen in Figure 4.

The multivariate analyses carried out in the RUDL condition did not show a significant main effect of the time factor or a significant interaction of time x group. The differences between pairs can be seen in Figure 5.

Meanwhile, a significant main effect of the time factor was found on the results for proprioception (F_1,18_ = 11.53, *p* < 0.003, η^2^*p* = 0.39). It should be noted, though, that just as in the previous postural control conditions there was no significant interaction between group x time. The measurements of the scores on the cognitive task performed during the proprioception tests did not show a significant main effect of the time factor or interaction. Differences between pairs of proprioception variables can be observed in Figure 6.

## 4. Discussion

The aim of this study was to evaluate the effects of an eight-week training program on the postural control and knee proprioception of healthy adults, with the inclusion of a cognitive task performed simultaneously. To accomplish this, both groups carried out motor training based on postural control and proprioceptive exercises, including in the experimental group a musical base that they had to hum or sing during the training to include a cognitive input to the motor training. The most notable results are that the training, despite having a positive effect on the results of the post-intervention motor tests (postural control and proprioception), has no effect on the cognitive task. Furthermore, there are no differences in the results between the control and experimental groups, so both training sessions produce the same effect in the subjects who perform it.

The main finding shows that a motor training program aimed at improving motor capacity has a positive impact on the results of proprioception and postural control tests after eight weeks of training. That is, the scores on the proprioception and postural control tests with eyes open and closed, with a simultaneous cognitive task, improved both in the EG (who performed the motor task with sung music) and in the CG (who performed the same motor task without musical accompaniment). In light of the results, it seems that the intervention with the inclusion of a cognitive task such as singing unknown music did not pose a greater cognitive processing challenge than the one posed to the control group, which only had verbal reinforcement from the researcher during the execution of the same exercises. A plausible elucidation for the observed lack of disparities between the experimental and control cohorts may reside in the fact that, in the absence of musical stimuli, the investigator endeavored to identify alternative methodologies to enhance the motivational levels of the participants within the control group. Such endeavors aimed at invigorating the members of the control group could have exerted a beneficial impact on their attentional focus. It is widely acknowledged within the academic community that augmenting the attentional capacity of participants serves to enhance the conscious assimilation of perceptual–motor attributes, encompassing proprioceptive awareness and postural regulation [42,43]. This, in turn, may have contributed to a more uniform performance across both groups, thereby blurring the distinctions that might otherwise have been evident in the presence of musical cues. The motivational intensity of the researcher may have been a contaminating factor in the study, since it could have served as a cognitive stimulus for the CG and might have thus contributed to the lack of differences between the scores on the cognitive task of the EG with respect to those of the CG. In other words, it is possible that both groups trained indirectly under the same motor–cognitive dual task paradigm. In this sense, these results would be in line with studies that show that dual training plays an important role in the transfer of the skill worked on, so specific training of motor–cognitive skills can favor positive results in the analyzed tests. Such training methods have also been found to have durability over time [30,44].

Another possible explanation for the main finding could be related to the kinds of athletic activities that the participants had carried out prior to the study. The members of both groups had participated in directed choreographed activities with musical accompaniment in their prior training programs with a frequency of two to three sessions per week. This could have interfered with the results, as the cognitive task of singing unfamiliar music when exercising may not have represented as great a difficulty for them. In other words, there may have been no additional challenge beyond what they were used to, even if it was a different activity than usual [45].

It is also possible that the characteristics of the proposed training, featuring the use of a fixed attentional focus for the motor task and a secondary focus on the cognitive task, could have contributed to improving the participants’ results of the motor–cognitive tests. However, no differences were found between the group with cognitive tasks and the one without. In line with what was mentioned above, the variable prioritization of the parameters to be trained, that is, the motor and cognitive tasks developed in the training, is considered essential for the effectiveness of dual training [44], since it has been demonstrated that, in order for dual training to be effective at improving specific variables, it is necessary to give instructions that place a priority on these specific variables or skills. In other words, instructors must change the attentional focus of the motor and cognitive task sequentially [46,47].

It should also be noted that few studies have specifically used music in training programs aimed at healthy and physically active subjects. This means that there is still little data in this regard, and a greater number of interventions are needed to paint a more complete picture of how music can affect performance on motor exercises. However, those authors who have introduced multitasking exercises based on music and rhythm in older adult populations have found that the available cognitive resources are increased, thus improving gait control [48] and balance, as well as producing a reduction in the rate of falls in said population [29]. Furthermore, dance or choreographed workouts with music have been shown to be a therapeutic tool for people with cognitive and motor pathologies such as Parkinson’s, cancer, or neurological disorders [49,50,51], as these activities include movements of all parts of the body and appear to improve adherence to sports activities, as well as leading to benefits in cognitive and balance parameters [50,51]. At the same time, activities with music have a strong social and motivational factor that favors the improvement of the well-being of subjects’ lives and contributes to the fight against a sedentary lifestyle in people with previous pathologies [52,53]. Although these authors have shown that music, dance, and rhythm produce better results, there are no studies that have used music as a cognitive challenge, and in our case, we did not focus on the rhythm but rather on the subjects learning the lyrics and memorizing them.

Considering the results, it seems that the simple physical training program of the perceptual–motor qualities that was carried out here led to improvements in the results of the postural control tests with open and closed eyes, as well as the results for knee proprioception. As confirmed by various authors, training programs that are aimed at single tasks (in this case, motor tasks) and that work on parameters similar to those used in the measurement tests tend to be an effective way to improve the results of said tests, due to automation and transfer of the tasks performed. Additionally, tasks that include explicit instructions that focus on postural control when training under dual tasks also improve motor results [30,54,55]. That is, training in specific concepts through frequent repetition of exercises replicating the specific tasks to be measured tends to improve performance [56]. Furthermore, it is known that focusing training on physical qualities such as proprioception and postural control delays the aging process, linked to the fragility of older adults, and prevents the risk of falls [54,57].

Within the training program, both groups performed exercises where postural control and balance were compromised in positions in which the knee oscillated between 45° of flexion and full extension. It is accepted that the lower limbs, and especially the knee joint, play a very important role in postural control. The position of the knee, along with the position of the hip, ankle, and toes, provides greater stability to the body. This stability is achieved thanks to the continuous flow of proprioceptive information through the ligaments, joint capsules, and muscles that surround these joints, which allow anteroposterior stabilization and rotation of the knee [58]. In turn, the 45° flexion position has a direct relationship with dynamic and static balance, since it is the position from which the ability to generate force in the quadriceps and hamstrings increases. These muscles are essential for motor skills and stability of the body [59].

When it came to the effects of the training program on unipodal postural control, no significant differences were found between the pre-test and the post-test. Indeed, there was even a slight worsening of motor performance on the second test, although it was not significant. It is possible that adding the auditory afferent of sung music in the experimental group may have worsened their conscious registration of the afferents related to balance, as well as of the proprioceptive afferents that could improve postural control and help maintain balance. This may have caused the results of the unipodal postural control tests to remain unchanged or to have worsened slightly after the intervention. This may be due to the inability of the nervous system (NS) to process a sensory input of a motor nature in the primary sensory cortex when the input is at a lower intensity and frequency than the auditory input, since the auditory and motor neuronal pathways and their brain processing area are closely linked [60]. The present results are in line with previous studies, as it has been shown that older adults make use of higher-level sensorimotor cortical areas during complex motor tasks (such as unipodal postural control), which may cause a greater dependence on cognitive information to process cortical sensory information that allows for controlling movement [61]. It is also necessary to mention that both the difficulty of the motor task and the reduction in the support base when performing the exercises and motor tests can cause the results to vary, producing a greater displacement of the CoP [62,63]. When there is a reduction in the base of support, there is a higher requirement for proprioceptive information to align the joints as well as possible on the point of support, causing vision, touch, or vestibular information, which are not the only ones who interfere [64].

There were no significant differences between the pre-test and post-test results of the cognitive task, which means that the subjects did not change the number of responses given to the semantic fields proposed in the “categorical evocation in associations test”, which was analyzed simultaneously with the motor tests. This may have several explanations, the first of which would be the possible adaptation by the subjects to the tests, given that the semantic fields proposed for the analysis of the cognitive task were not modified [30]. Furthermore, the fact that the cognitive training carried out had no direct relationship with the cognitive test may have influenced the degree of neuroplasticity created in the brain. When more specific exercises are worked on repetitively, this can lead to a greater change in test performance [43,65]. Additionally, the characteristics of the subjects in the sample may have influenced their performance on the cognitive test, since they had a high cultural level, as shown by the results on the CR scale. This seems to indicate that the cultural background of the subjects helped them to execute the double cognitive task, since the higher the cultural level of a subject, the greater the formation of neural networks that allow them to associate new learning with prior learning, thus improving cognitive functions and reducing their deterioration [66,67].

There is controversy when it comes to determining the duration and type of dual task training programs that improve the motor performance of healthy older adult subjects. Some researchers claim that in order to produce improvements one needs between 8 and 12 weeks of two to three sessions of 15 to 45 min per week [24]. Others consider a minimum of 12 weeks of dual task training to be necessary for the effect to be maintained over time, and they argue that variable-priority instructions must be given in motor–cognitive tasks [55]. A recent systematic review highlights that the main limitation of dual task training programs is the heterogeneity in the exercises carried out and the lack of specification of the difficulty of the secondary task [30]. Despite the variations among researchers, the results of this article indicate that upcoming studies should explore an 8-week motor training program, consisting of two sessions each week, as it enhances motor skills like postural control and proprioception. Therefore, integrating exercises that mimic everyday movements in these sessions may produce beneficial results in the assessments conducted. Nonetheless, it should be emphasized that in our research, the use of music did not influence cognitive performance. For subsequent investigations, it will be essential to utilize cognitive tasks that are closely related to the desired outcomes we intend to assess, ensuring they offer an appropriate challenge for participants while considering the unique characteristics of the sample.

Although there is scientific evidence on the consequences of the cognitive–motor deterioration suffered by the older adult population [30], there are few studies in populations of healthy and physically active older adults, so more studies are necessary to allow for the creation of training programs suitable for this population, especially given increases in longevity in a healthy society.

## 5. Conclusions

The development of an eight-week physical training program aimed at improving proprioception and postural control has improved performance on postural control (with open eyes and closed eyes) and proprioceptive tests but has not improved control results for unipodal posture. The dual cognitive–motor task intervention with the inclusion of unknown sung music in the EG training sessions did not improve performance in the postural control and proprioception tests carried out with a simultaneous cognitive task with respect to the CG. More research is needed on dual training in healthy and physically active subjects to adapt training with dual motor–cognitive tasks to the needs of this population group.

## Figures and Tables

**Figure 1 healthcare-12-02297-f001:**
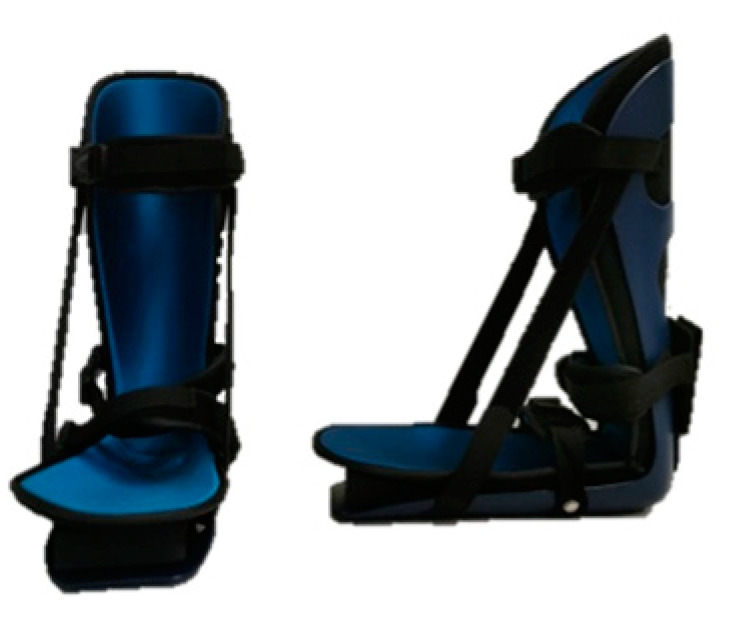
The orthopedic boots used in the study.

**Figure 2 healthcare-12-02297-f002:**
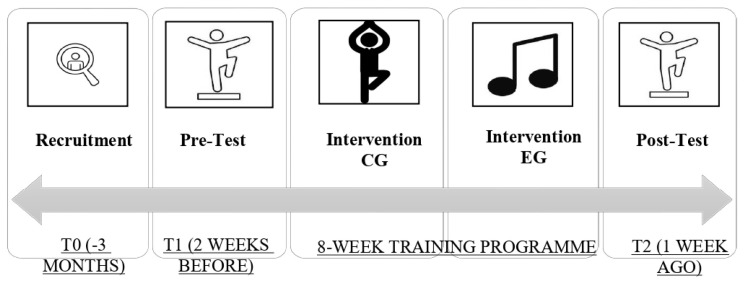
Timeline of protocol design.

**Figure 3 healthcare-12-02297-f003:**
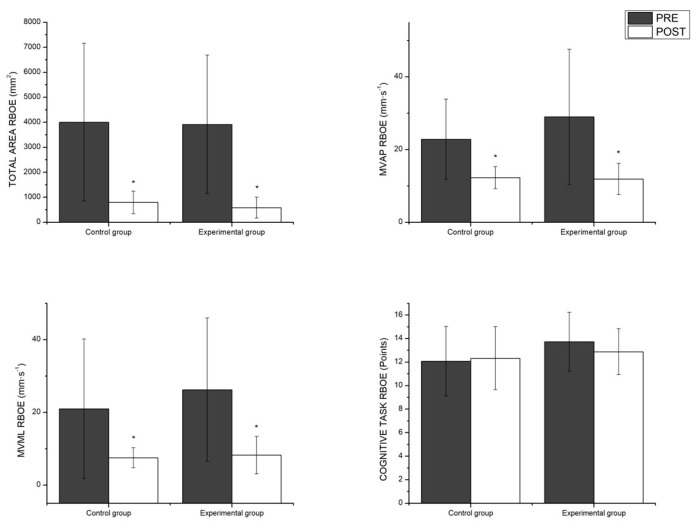
Pairwise comparisons of the group factor as a function of time in the open eyes postural control variables. RBOE = Romberg bipodal open eyes; MV_AP_ = mean velocity anterior–posterior; MV_ML_ = mean velocity medio–lateral. * significant main effect of the time factor; *p* < 0.05.

**Figure 4 healthcare-12-02297-f004:**
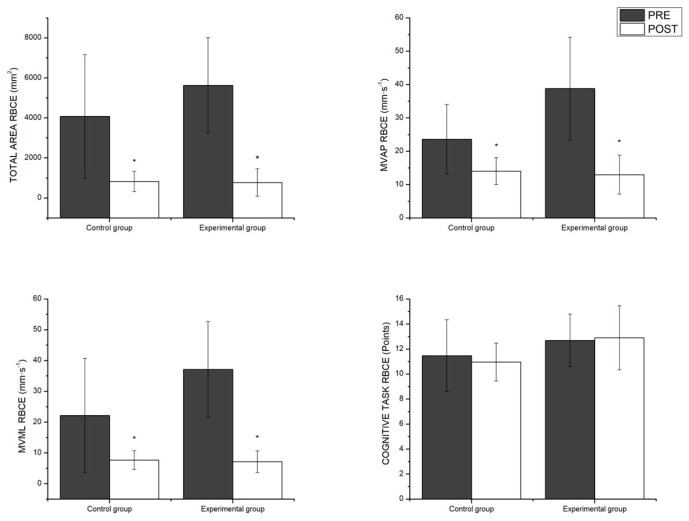
Pairwise comparisons between groups as a function of time for the postural control and eyes closed variables. RBCE = Romberg bipodal closed eyes; MV_AP_ = mean velocity anterior–posterior; MV_ML_ = mean velocity medio–lateral. * significant main effect of the time factor; *p* < 0.05.

**Figure 5 healthcare-12-02297-f005:**
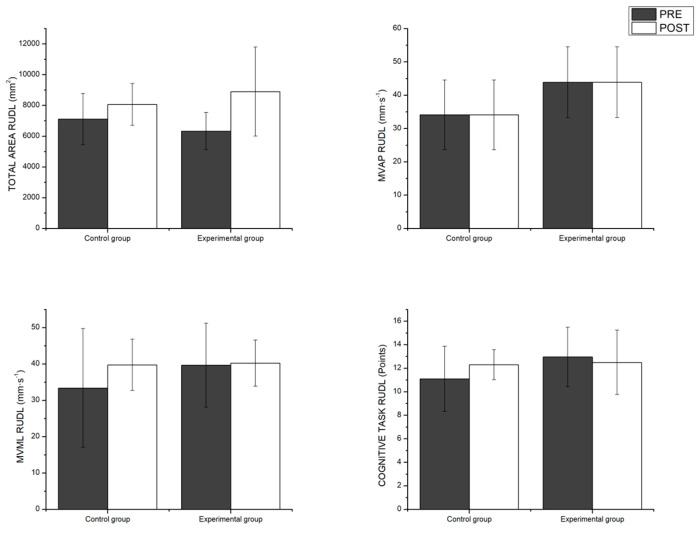
Pairwise comparisons between groups as a function of time variables measuring unipodal postural control with the dominant leg. RUDL = Romberg unipodal dominant leg; MV_AP_ = mean velocity anterior–posterior; MV_ML_ = mean velocity medio–lateral.

**Figure 6 healthcare-12-02297-f006:**
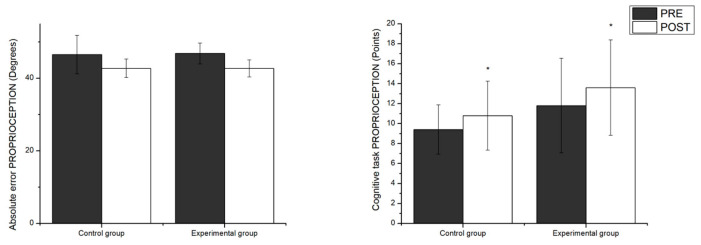
Pairwise comparisons between groups as a function of time for proprioception. * significant main effect of the time factor; *p* < 0.05.

## Data Availability

The data can be found at the following link: https://doi.org/10.6084/m9.figshare.25982335.v1.

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
