# Peer review of "The Effects of an 8-Week Cognitive–Motor Training Program on Proprioception and Postural Control Under Single and Dual Task in Older Adults: A Randomized Clinical Trial"

_healthcare, 2024, doi:10.3390/healthcare12222297_

Round 1
Reviewer 1 Report
Comments and Suggestions for Authors
Thanks to the authors for this study.
The TITLE of the study is inclusive and summarizes the content.
The general outlines of the study are adequately mentioned in the ABSTRACT section.
The language of the work is clear, understandable, and fluent. I did not encounter any language editing requirements.
The INTRODUCTION is long and comprehensive. In general, authors write about the subject from general to specific and then emphasize the relevant points sufficiently by turning towards the target. The objective and hypothesis are clear and precise.
In the MATERIALS METHODS section, all the steps and applications of the study are explained in detail and supported with visuals where necessary. The important steps are specified in the statistics section.
The RESULTS section is very successful. the findings are supported with visuals. it is written in a way that complements rather than repeats each other.
The length of the DISCUSSION is enough. Current literature is mentioned. important points are emphasized. possible explanations are given.
My only constructive criticism would be to have an introductory paragraph summarizing the purpose of the study, why it is important, and the most important findings in general, instead of writing the purpose exactly in the first paragraph of the discussion.
Overall thanks to the authors
Author Response
RESPONSE TO REVIEWRS
Article title: The effects of an 8-week cognitive-motor training program on proprioception and postural control under single and dual-task in older adults: a randomized clinical trial.
Dear editor and reviewers:
The authors of the study greatly appreciate the comments received from the editor and the four reviewers, which have helped us to be clearer in our writing and more precise in the presentation of the introduction, methods, results and discussion. We hope to have been able to provide a satisfactory response to all your comments, and we consider that the article has improved thanks to your contributions. If not, please do not hesitate to contact us again.
Below, you will find the response to each of the comments in italics explaining how the article has been revised according to the comment, and where the change in the article is located, to facilitate the reviewer's task. All changes made to the article are marked in yellow for better identification.
REVIEWER 1
Comments and Suggestions for Authors
Thanks to the authors for this study.
Comment 1: The TITLE of the study is inclusive and summarizes the content.
Response 1: Thanks for your comment
Comment 2: The general outlines of the study are adequately mentioned in the ABSTRACT section.
Response 2: Thanks for your comment
Comment 3: The language of the work is clear, understandable, and fluent. I did not encounter any language editing requirements.
Response 3: Thanks for your comment
Comment 4: The INTRODUCTION is long and comprehensive. In general, authors write about the subject from general to specific and then emphasize the relevant points sufficiently by turning towards the target. The objective and hypothesis are clear and precise.
Response 4: Thanks for your comment
Comment 5: In the MATERIALS METHODS section, all the steps and applications of the study are explained in detail and supported with visuals where necessary. The important steps are specified in the statistics section.
Response 5: Thanks for your comment
Comment 6: The RESULTS section is very successful.Tthe findings are supported with visuals. it is written in a way that complements rather than repeats each other.
Response 6: Thanks for your comment
Comment 7: The length of the DISCUSSION is enough. Current literature is mentioned. important points are emphasized. possible explanations are given.
Response 7: Thanks for your comment
Comment 8: My only constructive criticism would be to have an introductory paragraph summarizing the purpose of the study, why it is important, and the most important findings in general, instead of writing the purpose exactly in the first paragraph of the discussion.
Response 8: We appreciate your comment, and based on it we have expanded the introductory paragraph in the discussion section (pag 9 line 362-371)
Comment 9: Overall thanks to the authors
Response 9: Thank you very much for your constructive comments on the work done

Reviewer 2 Report
Comments and Suggestions for Authors
This study aimed to compare the effects of two types of motor-cognitive training with and without music on balance and proprioceptive function of the knee in elderly individuals. The results showed that both types of exercises had a positive and similar effect on balance and proprioceptive function but had no impact on cognitive-motor performance. The main points to note in this study are the inclusion of additional information in the introduction and the lack of sufficient justification for conducting this study compared to previous research. Additionally, some research variables and their relationship to balance were not properly explained. In the methodology, the implementation method of the exercise protocols and tests, as well as their number, are not appropriate. The results of this study (as reported in the discussion) were influenced by the participants' prior training experiences. Main effect tables and interaction between factors tables, as well as tables related to the participants' characteristics, were not provided. In the discussion and throughout the manuscript, there is insufficient and untimely referencing. The points mentioned in the discussion mostly emphasize that the participants had training backgrounds that affected their performance in this study and that an appropriate cognitive test was not selected. For these reasons, this article is not suitable for publication in journal if Health Care.
L69: Up to this point, there has been discussion about aging, single-task, and dual-task exercises, which could be removed without significantly affecting the article. It is suggested that these topics be presented in a single paragraph.
L74: Do proprioceptive exercises effectively reduce bone fractures? This sentence is misleading and confusing for the reader. Additionally, the two references provided (17 and 18) are not related to the proprioceptive system.
L76: It is mentioned that "Work on this abilities, has been demonstrated to reduce the risk of falls in older people." Which abilities do the authors refer to?
L84: Previous studies have reported extensively on the positive effects of balance exercises, dual-task exercises, and exercises accompanied by music on balance and cognitive functions. What sets this research apart from similar studies?
L96-101: The rationale for conducting research on healthy elderly individuals should be explained. As mentioned in line 98, there are numerous studies in the field of healthy aging, such as:
https://doi.org/10.3389/fnagi.2021.589299
https://link.springer.com/article/10.1186/s12877-020-1484-5
https://peerj.com/articles/15030/
https://jamanetwork.com/journals/jamainternalmedicine/article-abstract/226932
L103: In this study, quality of life was not assessed. So why discuss a variable that was not evaluated by this study?
The problem statement is unclear and contains many extraneous details, while the variables and reasons for repeating a study with many similar precedents are not mentioned. Variables such as the proprioceptive system in the knee or quality of life are suddenly introduced into the problem statement without proper context.
L107: The problem statement does not provide sufficient reasons regarding the impact of balance exercises on the proprioceptive system, nor does it establish a proven link between these two variables. Then, one of the objectives of this paper is suddenly stated to examine the effect of motor-cognitive exercises on these receptors, specifically in the knee joint. Why this joint? Is there a connection between motor-cognitive exercises and proprioceptive receptors? Unfortunately, the problem statement is not well-structured.
L110-111: The abstract states that both groups had similar motor-cognitive exercises, and only the experimental group used music during the exercises.
L118: What were the criteria for selecting the number of participants?
L120: No explanation has been provided regarding CR at all.
L150: For the use of orthopedic boots and their evaluation, please provide a reference.
L160: What was the rest time between each set of three repetitions?
L175-176: Why was it necessary to standardize the single-leg balance test with the proprioceptive test?
The Romberg test has specific guidelines and execution methods. Why did you use a boot for this purpose? Please provide a reference for this matter.
L177: Between each repetition, how much rest time did you allow? How much rest time was there between each test?
L186-187: This sentence is repeated in line 177.
L189: In this section, the following questions arise which are not explained in the text:
Was the cognitive test evaluated on its own to allow for an appropriate comparison?
Does the balance test not get affected by the execution of the cognitive test, which requires speaking?
Please provide at least two references that have used this test as a dual task.
L202-204: How many tests did you have in total? First, state the number of tests, then provide the explanations. The presentation of the method is causing confusion for the reader.
Based on the explanations, there should be six tests.
L217: What score was used as the criterion for selecting participants from these two tests and why? Based on which reference did you choose the cut-off point?
L235-237: The presentation of motor exercises lacks references and is not clear.
L241: Is there a reference for implementing this exercise protocol? What was the intended effect of "memorizing song lyrics and singing them while performing physical movements"? Was this exercise meant to create a cognitive load? Why were the movements not performed in coordination with music?
L248: The appendix was not provided. The explanations regarding the exercise implementation are unclear. Performing single-person balance movements is likely more difficult than two-person movements due to the need to maintain balance. The explanations are insufficient, and no reference is provided for further consultation.
L256-257: The exercises focused on balance and maintaining a 45-degree angle in the knee. Is improvement in motor tests not predictable?
L261-263: This has been mentioned in previous sections and is repetitive.
L272: The balance test could also have been defined as another within-group factor with three levels: eyes open, eyes closed, and single-leg. This would reduce the error arising from the number of tests.
L278: Initially, a table should be provided detailing the characteristics of the participants. Were the participants male or female? What were the questionnaire scores that served as the entry criteria for each group, and other relevant information?
L280: The main effect of time is not for the groups. Line 282 shows the interaction; this phrase should not be here.
L283: The statistical test used for this analysis was not mentioned. Please provide the rationale for conducting this test in the statistical methods section.
L296: When there is no significant interaction between group and time, it means that the exercises had a similar effect in both groups, not different effects.
L304: Was there no significant difference between the two groups before the exercises?
L327: The terms "motor skills" or "physical training" are new and not relevant to the variables of this study.
L330: Single-task motor exercises showed improvement, not dual tasks.
L338-339: These instructions were not mentioned for the control group in the methods.
L351-352: This aspect was controllable and surely had an impact on your research.
Author Response
RESPONSE TO REVIEWRS
Article title: The effects of an 8-week cognitive-motor training program on proprioception and postural control under single and dual-task in older adults: a randomized clinical trial.
Dear editor and reviewers:
The authors of the study greatly appreciate the comments received from the editor and the four reviewers, which have helped us to be clearer in our writing and more precise in the presentation of the introduction, methods, results and discussion. We hope to have been able to provide a satisfactory response to all your comments, and we consider that the article has improved thanks to your contributions. If not, please do not hesitate to contact us again.
Below, you will find the response to each of the comments in italics explaining how the article has been revised according to the comment, and where the change in the article is located, to facilitate the reviewer's task. All changes made to the article are marked in yellow for better identification.
REVIEWER 2
This study aimed to compare the effects of two types of motor-cognitive training with and without music on balance and proprioceptive function of the knee in elderly individuals. The results showed that both types of exercises had a positive and similar effect on balance and proprioceptive function but had no impact on cognitive-motor performance. The main points to note in this study are the inclusion of additional information in the introduction and the lack of sufficient justification for conducting this study compared to previous research. Additionally, some research variables and their relationship to balance were not properly explained. In the methodology, the implementation method of the exercise protocols and tests, as well as their number, are not appropriate. The results of this study (as reported in the discussion) were influenced by the participants' prior training experiences. Main effect tables and interaction between factors tables, as well as tables related to the participants' characteristics, were not provided. In the discussion and throughout the manuscript, there is insufficient and untimely referencing. The points mentioned in the discussion mostly emphasize that the participants had training backgrounds that affected their performance in this study and that an appropriate cognitive test was not selected. For these reasons, this article is not suitable for publication in journal if Health Care.
First of all, we want to thank you for the exhaustive review you have done to the article, which undoubtedly adds quality to the article. We have tried to respond to all the comments you have made to us and make the appropriate changes so that it is in agreement with all the reviewers. We hope that after having gathered the contributions of all the reviewers, the article will be ready to be published in this journal.
Comment R2: L69: Up to this point, there has been discussion about aging, single-task, and dual-task exercises, which could be removed without significantly affecting the article. It is suggested that these topics be presented in a single paragraph.
Response 1:We appreciate your comment, but there are other reviewers who have asked to increase the content of what was mentioned in the introduction regarding these concepts that you mention, and we consider that to contextualize them it is necessary to maintain it.
Comment R2: L74: Do proprioceptive exercises effectively reduce bone fractures? This sentence is misleading and confusing for the reader. Additionally, the two references provided (17 and 18) are not related to the proprioceptive system.
Response 2: We appreciate your comment and affirm that it was a writing error to talk about bone fractures, we have modified the paragraph and we have updated the bibliography, I can see it on page 2 line 90-94.
Comment R2: L76: It is mentioned that "Work on this abilities, has been demonstrated to reduce the risk of falls in older people." Which abilities do the authors refer to?
Response 3: We appreciate your comment and after reviewing the previous comment we have modified it, since we are not referring to skills but to capabilities such as postural control and proprioception, and thanks to physical-cognitive training, these capabilities are improved and thus avoid falls.
Comment R2: L84: Previous studies have reported extensively on the positive effects of balance exercises, dual-task exercises, and exercises accompanied by music on balance and cognitive functions. What sets this research apart from similar studies?
Response 4: We appreciate this question since we will be able to explain to you that despite there being previous studies like those presented here, many of them have a sample of older people with cognitive pathologies. Furthermore, several articles suggest that the subjects are not randomized into the different groups but rather that they themselves choose where to participate. In our case there is no choice and regardless of their characteristics they perform one or the other training. At the same time, and once the study is completed, we consider that the fact of having introduced the CR as a measurer of the cultural level of the subjects, lets us know that we have a peculiar group, physically very active and with a high cultural level, a fact that In hindsight we will see that it limits the results. We also consider that the fact of measuring proprioception together with cognitive tasks has not been carried out in many studies, since the majority measure gait, isolated balance, which we believe is important to take into account, since it has been shown that the The fact of training proprioception improves the performance of postural control and intrinsically this favors the delay in the appearance of injuries and the improvement of the quality of life of the subjects at these ages (Lee et al. 2015; Nieto-Guisado et al., 2022). Regarding the inclusion of music, in our case we do not use it as a rhythmic base but as a cognitive stimulus, we do not follow the rhythm of the music but we intend for the subjects to learn the lyrics, creating a cognitive challenge for this. We consider that these are the factors or details that make the reason for carrying out this study different.
Comment R2: L96-101: The rationale for conducting research on healthy elderly individuals should be explained. As mentioned in line 98, there are numerous studies in the field of healthy aging, such as:
https://doi.org/10.3389/fnagi.2021.589299
https://link.springer.com/article/10.1186/s12877-020-1484-5
https://peerj.com/articles/15030/
https://jamanetwork.com/journals/jamainternalmedicine/article-abstract/226932
Response 5: We appreciate the different articles provided on the subject, and based on some of them and others previously mentioned, we have expanded the information on why to carry out studies of this type in a healthy adult population, which can be found on page 3 line 116-122.
Comment R2: L103: In this study, quality of life was not assessed. So why discuss a variable that was not evaluated by this study?
Response 6: We appreciate your comment. We believe that despite not measuring quality of life specifically and not being the specific object of study, implicitly when we carry out this type of motor-cognitive programs, we do so with the aim of improving the quality of life of people as they are. As the scientific literature says, it is a justification for working with this population group.
The problem statement is unclear and contains many extraneous details, while the variables and reasons for repeating a study with many similar precedents are not mentioned. Variables such as the proprioceptive system in the knee or quality of life are suddenly introduced into the problem statement without proper context.
Comment R2: L107: The problem statement does not provide sufficient reasons regarding the impact of balance exercises on the proprioceptive system, nor does it establish a proven link between these two variables. Then, one of the objectives of this paper is suddenly stated to examine the effect of motor-cognitive exercises on these receptors, specifically in the knee joint. Why this joint? Is there a connection between motor-cognitive exercises and proprioceptive receptors? Unfortunately, the problem statement is not well-structured.
Response 7: We appreciate your comment, due to the length of the text we have had to cut out a lot of information like this one that you point out to us, but to respond to it, we have modified many parts of the introduction adding information about it, even reformulating the problem statement for greater clarity, hopefully that is more appropriate now, if not, let us know so we can modify the text again and add more information about the role that proprioception plays in postural control. These same authors have published a previous article talking about the relationship between postural control and proprioception (Nieto-guisado et al., 2022).
Comment R2: L110-111: The abstract states that both groups had similar motor-cognitive exercises, and only the experimental group used music during the exercises.
Response 8: We appreciate your comment, and after reviewing the summary, we do not find that we have stated that both groups perform similar motor-cognitive training, but rather we specify that both groups perform the same motor training but that the experimental group includes the inclusion of sung music. in their sessions unlike the control group.
Comment R2: L118: What were the criteria for selecting the number of participants?
Response 9: We appreciate your question and take this opportunity to clarify that there is no criterion for selecting the sample, but as explained in the text (page 3 line 152-153), it is a convenience sample since it was announced by different sports centers where They taught classes for the elderly group and were the ones who signed up to voluntarily participate in the program. If the reviewer considers it appropriate, we can perform a post-hoc statistical power analysis and report it in the sample description section.
Comment R2: L120: No explanation has been provided regarding CR at all.
Response 10: We appreciate the appreciation and have added more information in the introduction about it, page 2 line 75-85.
Comment R2: L150: For the use of orthopedic boots and their evaluation, please provide a reference.
Response 11: We provide a reference that deals with the use of ski boots and their functionality (Noé et al., 2020) number 35, and we explain that we replaced the ski boot with orthopedic ones given their ease of putting on and taking off, being able to have the same for everyone, since a ski boot costs a lot of money and because older people might feel uncomfortable wearing them.
Comment R2: L160: What was the rest time between each set of three repetitions?
Response 12: We appreciate the appreciation and have included an explanation on page 5 line 208-2010 since it was not understood correctly.
Comment R2: L175-176: Why was it necessary to standardize the single-leg balance test with the proprioceptive test?
Response 13: We appreciate your question and considering that you refer to why we only did the dominant leg in both tests, it was to lighten the testing process for older people since we consider that the process had a long duration, and this test requires a higher concentration due to risk of falls. By selecting only the dominant leg, it seemed correct to us that both tests had the same conditions and hence both the single-leg Romberg test and proprioception were evaluated in the dominant leg.
Comment R2: The Romberg test has specific guidelines and execution methods. Why did you use a boot for this purpose? Please provide a reference for this matter.
Response 14: We believe that the reference you are referring to is added on page 4 line 180.
Comment R2: L177: Between each repetition, how much rest time did you allow? How much rest time was there between each test?
Response 15: As we mentioned previously, between each repetition there was 30 seconds of rest and between test and test 3 minutes, to give the subject time to get off the platform, explain the next test and begin. We are introducing this information on page 5 line 219-220.
Comment R2: L186-187: This sentence is repeated in line 177.
Response 16: We appreciate your comment and apologize since the information was poorly detailed.
Comment R2: L189: In this section, the following questions arise which are not explained in the text:
Was the cognitive test evaluated on its own to allow for an appropriate comparison?
Does the balance test not get affected by the execution of the cognitive test, which requires speaking?
Please provide at least two references that have used this test as a dual task.
Response 17: We appreciate the questions regarding the cognitive test and we must add that we have not currently found references that have used the Barcelona test as a dual task with motor tests. The reason for selecting this test was because we considered that the cognitive task could be an important stimulus for the profile of subjects we had and because we wanted to innovate in the cognitive test, since other studies use the Strop test, counting memorize words or numbers among others.
Comment R2: L202-204: How many tests did you have in total? First, state the number of tests, then provide the explanations. The presentation of the method is causing confusion for the reader.
Based on the explanations, there should be six tests.
Response 18: A total of three different tests were carried out, 1. Romber's Test; 2. Propioceptive Test; 3. Cognitive test, a subscale of the Barcelona Test called “categorical evocation in associations. The Romberg test has 3 conditions, a) Romberg Bipodal Open Eyes; b)Romberg Bipodal Closed Eyes; c) Romberg Unipodal Dominant Leg. The proprioceptive test has a single measurement condition, dominant leg, eyes closed. The conditions of both tests were repeated a total of 3 times each, that is, 3 attempts per condition and to each of them the simultaneous completion of the subscale of the Barcelona test was added. Each attempt lasted 30 seconds, with a 30-second rest period between attempts. Once all attempts of each test were completed, there was a 3-minute rest time. To this we can add the Cognitive Reserve scale and the IPAQ questionnaire that is given to the subjects prior to beginning the testing protocol.
Comment R2: L217: What score was used as the criterion for selecting participants from these two tests and why? Based on which reference did you choose the cut-off point?
Response 19: The sample, as we have previously pointed out, was of convenience, we asked that the inclusion criteria be met and there was no test for the selection of the participants. Despite passing the CR scale, this was not a determining factor in whether or not to include the participants in the sample, but rather it was used to respond to the analysis of the results.
Comment R2: L235-237: The presentation of motor exercises lacks references and is not clear.
Response 20: We appreciate the comment and hope that with the inclusion of an Annex (pag 6 line 285) that explains in detail the sessions carried out with their respective exercises, your question will be answered. Even so, we want to point out that the choice of these exercises is related to the postural control and proprioception tests carried out.
Comment R2: L241: Is there a reference for implementing this exercise protocol? What was the intended effect of "memorizing song lyrics and singing them while performing physical movements"? Was this exercise meant to create a cognitive load? Why were the movements not performed in coordination with music?
Response 21: We appreciate your questions and we will try to answer all of them as correctly as possible. There are articles, such as those mentioned in the text and those that you have provided us, that show motor training protocols with musical bases, working these through rhythm. In our case, and after thinking about how to include music not as a rhythmic task but as a cognitive load, it occurred to us, with the help of a psychologist, to introduce the lyrics of the songs as a cognitive task and not have the subjects follow the rhythm of the song. music was mandatory if not the lyrics were learned. It should be noted that the exercises performed when playing music, many of them included coordination with it, but it was not the main objective.
Comment R2: L248: The appendix was not provided. The explanations regarding the exercise implementation are unclear. Performing single-person balance movements is likely more difficult than two-person movements due to the need to maintain balance. The explanations are insufficient, and no reference is provided for further consultation.
Response 22: We appreciate your comment and apologize for not having been able to show the appendix, currently the detailed sessions are available as complementary material. We point out that the protocol is our own, it is not taken from other articles but from the thesis that was carried out and in which said article is included. Hopefully this is enough information and if not, please let us know so we can complete it
Comment R2: L256-257: The exercises focused on balance and maintaining a 45-degree angle in the knee. Is improvement in motor tests not predictable?
Response 23: We appreciate the comment and we agree with you, but it was what the authors were looking for, that there was a direct relationship between the tests measured and the exercises carried out so that, based on training, the subjects improved their motor abilities, also pointing out the role of proprioception. in improving postural control. At the same time, we emphasized that the objective was to know how the cognitive task influenced motor performance, which is why we wanted the proposed exercises to be as reliable as possible to the tests measured and the parameters that we wanted to improve.
Comment R2: L261-263: This has been mentioned in previous sections and is repetitive.
Response 24: We appreciate the comment and have modified the text (page 6 line 298-299).
Comment R2: L272: The balance test could also have been defined as another within-group factor with three levels: eyes open, eyes closed, and single-leg. This would reduce the error arising from the number of tests.
Response 25: The option of considering the levels of the balance test as an intragroup condition was a possibility that we were seriously considering. Finally we decided to analyze it separately, which is much simpler to explain, since otherwise we would have to analyze the data with an omnibus test and the explanation of the results would be more complicated.
Comment R2: L278: Initially, a table should be provided detailing the characteristics of the participants. Were the participants male or female? What were the questionnaire scores that served as the entry criteria for each group, and other relevant information?
Response 26: We appreciate your comment and we have expanded the information about the participants described in section 2.2 (line 149-152). Due to the word and text limit of the article, we have preferred not to add a table due to the similarity between both groups.
Comment R2: L280: The main effect of time is not for the groups. Line 282 shows the interaction; this phrase should not be here.
Response 27: First, the significant of the main effect of time factor (pre and post, no for the groups) is reported, indicating that there are significant differences between the pre and post scores in both groups. Next, the effect of the time x group interaction is reported, which is not significant, and it is explained that the pre and post differences are not caused by being in the control or experimental group, as both groups improve.
Comment R2: L283: The statistical test used for this analysis was not mentioned. Please provide the rationale for conducting this test in the statistical methods section.
Response 28: The statistical tests are explained in the statistical methods section on lines 301-313.
Comment R2: L296: When there is no significant interaction between group and time, it means that the exercises had a similar effect in both groups, not different effects.
Response 29: We greatly appreciate your review work, obviously there is an error in the wording and the sentence should be formulated in the negative since if there is no significant effect on the interaction the groups will have had similar results. We have corrected the wording as follows: "no significant group x time interaction effect was found, so the intervention program had no differing effects as a function of the group" (pag 7 line 334).
Comment R2: L304: Was there no significant difference between the two groups before the exercises?
Response 30: To verify that there were no differences between the groups in the initial phase, a t-student test was applied, which showed no differences. We have not reported it but we have currently added it to the beginning of the results section, page 7 line 318-319.
L327: The terms "motor skills" or "physical training" are new and not relevant to the variables of this study.
Response 31: We are changing the terms in the text.
Comment R2: L330: Single-task motor exercises showed improvement, not dual tasks.
Response 32: We apologize since we did not find the reference you make in this comment, but what we explain is that the dual tests (motor-cognitive) improve in both groups regardless of the intervention carried out.
Comment R2: L338-339: These instructions were not mentioned for the control group in the methods.
Response 33: We appreciate your comment and affirm that said information does not appear in the method since they were not instructions for the group, but when analyzing the results and evaluating the interventions we believe that it may be a limitation found in the study. We point out that both groups performed the same exercises, one including music in the sessions and others not, and that all sessions were carried out by a trainer who focused on the subjects performing the motor exercises correctly. If you consider that we should indicate the total intervention of the coach in the method, let us know and we will include it later.
Comment R2: L351-352: This aspect was controllable and surely had an impact on your research.
Response 34: As you point out, the fact that the subjects performed choreographed interventions may have interfered with our results, as we pointed out, but we highlighted that our interventions did not go to the rhythm of the music, but rather the subjects had an added cognitive task, which was music. We considered that we could not control this once the sample was selected and it was not an exclusion criterion for it.
Reviewer 3 Report
Comments and Suggestions for Authors
Abstract
Please provide quantitative data in the results section of the abstract (mean, SD, p-value, etc).
Introduction
I believe that when you talk about SN and CNS you are talking about the same thing so keep only one abbreviation throughout the text.
The second and third paragraphs involve important concepts about the field your research is inserted, but these concepts should be presented in a way that it gives background for the rationale for your study. The way it is sounds a little bit disconnect from your research. Try to integrate these concepts to justify your study and shorten your introduction.
Reference in a different style “…lower limbs (Riva et al., 2016), in addition…”
Methods
Please improve this description “…of between (73.25± 5.98 years) and with (56,8±16,78𝐶𝑅.” No need for parentheses to describe these data and the variables with mean±SD should be specified.
Was repositioning assessed with eyes closed? (page 4, line 154-163).
How were the different semantic fields chosen between tasks? Randomly? Were they graded in terms of difficulty?
Why medical history was not collected?
I think it is worthy including an experimental protocol with images and time points to illustrate the sessions and pre/post assessments for both groups.
I suggest including a table with demographic characteristics for both groups.
Results
What is the reason for improving proprioception points in both groups.
Discussion
Address the limitation of using “an application called 135 Goniometer Pro, installed on a smartphone Galaxy J7…”. An actual goniometer or a better tool would be idea to precisely measure range of motion.
Typo: “…necessary _for the effect to be…”
Author Response
RESPONSE TO REVIEWRS
Article title: The effects of an 8-week cognitive-motor training program on proprioception and postural control under single and dual-task in older adults: a randomized clinical trial.
Dear editor and reviewers:
The authors of the study greatly appreciate the comments received from the editor and the four reviewers, which have helped us to be clearer in our writing and more precise in the presentation of the introduction, methods, results and discussion. We hope to have been able to provide a satisfactory response to all your comments, and we consider that the article has improved thanks to your contributions. If not, please do not hesitate to contact us again.
Below, you will find the response to each of the comments in italics explaining how the article has been revised according to the comment, and where the change in the article is located, to facilitate the reviewer's task. All changes made to the article are marked in yellow for better identification.
REVIEWER 3
Abstract
Comment 1: Please provide quantitative data in the results section of the abstract (mean, SD, p-value, etc).
Response 1: We have expanded the information on the results in the abstract, but due to the word limit it is very difficult to report the results of the variables, we have chosen to indicate the significant effects and report the range in which the results of the variables have increased.
Introduction
Comment 2: I believe that when you talk about SN and CNS you are talking about the same thing so keep only one abbreviation throughout the text.
Response 2: We appreciate the comment and we have taken it into account and have unified the concept to central nervous system (CNS)
Comment 3: The second and third paragraphs involve important concepts about the field your research is inserted, but these concepts should be presented in a way that it gives background for the rationale for your study. The way it is sounds a little bit disconnect from your research. Try to integrate these concepts to justify your study and shorten your introduction.
Response 3: Based on your comment, we have made slight changes to the content of the text in order to better organize the content and make the concepts we discuss throughout the entire content of the article clearer and it currently appears in the text on the page 1,2 a line 40-69.
Comment 4: Reference in a different style “…lower limbs (Riva et al., 2016), in addition…”
Response 4: We appreciate your comment and we have made the modification.
Methods
Comment 5: Please improve this description “…of between (73.25± 5.98 years) and with (56,8±16,78??.” No need for parentheses to describe these data and the variables with mean±SD should be specified.
Response 5: We appreciate your comment and we have made the modification.
Comment 6: Was repositioning assessed with eyes closed? (page 4, line 154-163).
Response 6: We appreciate the question and the answer is yes, proprioception was assessed with eyes closed, since previous studies state that conscious repositioning should be measured with eyes closed to facilitate the training of proprioceptive sensory afferents (Goodman and Tremblay, 2018).
Comment 7: How were the different semantic fields chosen between tasks? Randomly? Were they graded in terms of difficulty?
Response 7: We appreciate the questions and the answers are that the semantic fields were chosen by the authors once the anamnesis and the cognitive reserve test were carried out, with which we were able to know their lifestyle and cultural level. Based on this, semantic fields were chosen that we considered could be known by all subjects and that, despite their difficulty, they could respond. The order of the fields in each task was selected at random, using the same criteria in the different tests. There was no classification based on the difficulty of the selected semantic fields and the tests since we did not want that choice to influence the development of the motor tests, so everything was random.
Comment 8: Why medical history was not collected?
Response 8: We appreciate the comment so that we can specify that although it is not stated literally, in the form that is sent to the participants to collect their personal data, questions do appear regarding their medical data, such as whether they have suffered or suffer from any cognitive disease, if they have a chronic illness or if they have any medical condition or prescription that prevents or makes it difficult for them to carry out the study. We considered not providing medical history, since a medical document as such was not requested, but rather that they themselves gave answers based on their medical history.
Comment 9: I think it is worthy including an experimental protocol with images and time points to illustrate the sessions and pre/post assessments for both groups.
Response 9: We appreciate your comment and we believe that the inclusion of a timeline in our study makes our article more enriching, which is why a figure has been added as an outline of the protocol carried out on page 6 line 259.
Comment 10: I suggest including a table with demographic characteristics for both groups.
Response 10: We appreciate your comment and we have expanded the demographic information of the entire group in the description of the sample by text length as other reviewers requested, you can view it on page 3 line 149-152.
Results
Comment 11: What is the reason for improving proprioception points in both groups.
Response 11: We appreciate the question and we consider just as postural control improved in both groups, the training for 8 weeks in sessions that integrated proprioceptive and postural control exercises implies an improvement in these qualities that are trained and practiced. In addition, there were specific exercises that replicated positions and movements similar to those measured in the proprioceptive tests, placing considerable emphasis on the ability to internalize specific positions of the lower body in order to prevent injuries or future falls in this population, making the when performing proprioceptive tests, subjects will have greater awareness of their joints in space and replicate positions more precisely and effectively.
Discussion
Comment 12: Address the limitation of using “an application called 135 Goniometer Pro, installed on a smartphone Galaxy J7…”. An actual goniometer or a better tool would be idea to precisely measure range of motion.
Response 12: We appreciate your appreciation and we take it into account for future studies, but in this case, although the use of a mobile app to measure proprioception may seem like a limitation of the study, for us this is not the case, since said application has been validated by previous authors and its results have been compared with results from traditional goniometers and has been considered reliable due to the degree of similarity between both measurements (Melian- Ortiz et al., 2019; Mourcou et al., 2016). Furthermore, consider that providing future authors with a free tool that is easy to use and integrated into a mobile phone, a device that we all have within reach, can be an advantage and help proprioception to be measured more recurrently, always with a prior training in the measurement protocol. Likewise, we know that there are instruments such as isokinetic dynamometry that are more precise and used, but they require an economic budget, facilities and services that we consider are not available to everyone.
Comment 13: Typo: “…necessary _for the effect to be…”
Response 13: We appreciate the note, and we have corrected the error.

Reviewer 4 Report
Comments and Suggestions for Authors
Keyword: Music
One-word music is unclear and not suitable.
Introduction
1. Inconsistent Citation Formatting: The citation formatting is inconsistent. For example, some references are cited within the text (e.g., "1,2”) while others are mentioned in parentheses (e.g., "(Riva et. al., 2016)" in line 78. A uniform citation style should be adopted for clarity and consistency.
2. Hypothesis Clarification: The hypothesis is mentioned briefly at the end of the introduction. However, it would be beneficial to clearly state the specific hypotheses of the study early on to set the stage for the research questions and objectives.
3. Motivation for Music: The introduction mentions the use of music in the training program but does not fully explain the rationale behind this choice. A more detailed explanation of why music was included and its expected effects would strengthen the introduction.
Materials and Methods
1. Adding Figure for Clarification: include the figure of protocol flowchart procedure for more clarification explanation
Result
1. Comparison of Groups: The study compares the experimental group (EG) with the control group (CG) and reports no significant differences between the groups. While this is clear, the discussion could be expanded to explore potential reasons for the lack of differences, particularly given the inclusion of music in the EG.
2. Adding justification as to why there are no significant differences.
3. Cognitive Task Performance: The study reports no significant differences in cognitive task performance post-intervention. It could benefit from a more detailed discussion of why this might be the case, considering the task's cognitive demands and the training program's potential effects.
Discussion
1. Comparison with Previous Studies: The discussion could include a more thorough comparison with previous studies to contextualize the findings. How do the results of this study align with or differ from existing research on cognitive-motor training in older adults?
2. Effect of Music: The study found no significant differences between groups with and without music. The discussion could delve deeper into why music did not have the expected effect and explore alternative explanations or hypotheses for future research.
3. Potential Biases: The discussion could address potential biases in the study, such as the motivational efforts of the researcher and the familiarity of participants with similar activities. Discussing how these biases might have influenced the results would strengthen the analysis.
4. Practical Implications: The discussion could explore the practical implications of the findings for designing training programs for older adults. For example, what specific components of the training program were most effective, and how can these be incorporated into real-world settings?
Comments on the Quality of English LanguageOverall, the document is well-written, but addressing these points would improve clarity, readability, and consistency, making it more accessible to a broader audience.
Maintaining consistency in terminology would improve clarity.
Grammatical error
-
Example: There are a few grammatical errors and typos that should be corrected. For example, "an 8-week cognitive-motor training program on postural control and knee proprioception under single and dual-task in older adults" should be "an 8-week cognitive-motor training program on postural control and knee proprioception under single and dual tasks in older adults."
Author Response
RESPONSE TO REVIEWRS
Article title: The effects of an 8-week cognitive-motor training program on proprioception and postural control under single and dual-task in older adults: a randomized clinical trial.
Dear editor and reviewers:
The authors of the study greatly appreciate the comments received from the editor and the four reviewers, which have helped us to be clearer in our writing and more precise in the presentation of the introduction, methods, results and discussion. We hope to have been able to provide a satisfactory response to all your comments, and we consider that the article has improved thanks to your contributions. If not, please do not hesitate to contact us again.
Below, you will find the response to each of the comments in italics explaining how the article has been revised according to the comment, and where the change in the article is located, to facilitate the reviewer's task. All changes made to the article are marked in yellow for better identification.
REVIEWER 4
Keyword: Music
Comment 1: One-word music is unclear and not suitable.
Response 1: We appreciate your comment, and we have changed music for training with music, hopefully it will be clearer and more concise.
Introduction
Comment 2: Inconsistent Citation Formatting: The citation formatting is inconsistent. For example, some references are cited within the text (e.g., "1,2”) while others are mentioned in parentheses (e.g., "(Riva et. al., 2016)" in line 78. A uniform citation style should be adopted for clarity and consistency.
Response 2: We appreciate the comment and apologize for the error, it has already been corrected and unified with the style requested by the magazine.
Comment 3: Hypothesis Clarification: The hypothesis is mentioned briefly at the end of the introduction. However, it would be beneficial to clearly state the specific hypotheses of the study early on to set the stage for the research questions and objectives.
Response 3: We appreciate your appreciation and we have taken it into account to reformulate the paragraph so that the objective we have with carrying out this work and the hypothesis we propose are clear. You can see the new text on pag 3 line 127-141.
Comment 4: Motivation for Music: The introduction mentions the use of music in the training program but does not fully explain the rationale behind this choice. A more detailed explanation of why music was included and its expected effects would strengthen the introduction.
Response 4: We appreciate your comment and that is why we have added more information about it on page 2,3 line 100-115. Even so, we consider that in the discussion we detailed more extensively the reason for using music in interventions with older people. If you think you need more justification in the introduction, let us know so we can expand it.
Materials and Methods
Comment 5: Adding Figure for Clarification: include the figure of protocol flowchart procedure for more clarification explanation
Response 5: We appreciate your comments and have added a timeline to make the protocol followed on page 6 line 259 more graphic.
Result
Comment 6: Comparison of Groups: The study compares the experimental group (EG) with the control group (CG) and reports no significant differences between the groups. While this is clear, the discussion could be expanded to explore potential reasons for the lack of differences, particularly given the inclusion of music in the EG.
Response 6: We appreciate your comment and and we consider that given the length of the article and the modifications already made, the content is quite explicit, if you consider that something more can still be added, let us know and we will introduce it later.
Comment 7: Adding justification as to why there are no significant differences.
Response 7: We appreciate your comment and consider that in the discussion section the reason for not having significant differences between groups in line x is justified; in the results it is detailed numerically and, in the discussion, referenced and explained.
Comment 8: Cognitive Task Performance: The study reports no significant differences in cognitive task performance post-intervention. It could benefit from a more detailed discussion of why this might be the case, considering the task's cognitive demands and the training program's potential effects.
Response 8: We appreciate your comment and we consider that given the length of the article and the modifications already made, the content is quite explicit, if you consider that something more can still be added, let us know and we will introduce it later.
Discussion
Comment 9: Comparison with Previous Studies: The discussion could include a more thorough comparison with previous studies to contextualize the findings. How do the results of this study align with or differ from existing research on cognitive-motor training in older adults?
Response 9: We have considered your comment and have included additional information in the text that you can see on page 9 line 380-390. Although, as we have previously pointed out, there are many articles that we mention to compare our results and we cannot go overboard.
Comment 10: Effect of Music: The study found no significant differences between groups with and without music. The discussion could delve deeper into why music did not have the expected effect and explore alternative explanations or hypotheses for future research.
Response 10: We have considered your comment and have included additional information in the text that you can see on page 10 line 416-432.
Comment 11: Potential Biases: The discussion could address potential biases in the study, such as the motivational efforts of the researcher and the familiarity of participants with similar activities. Discussing how these biases might have influenced the results would strengthen the analysis.
Response 11: We appreciate your appreciation of the biases that occurred during the study and we explain them in the discussion section page x line x, where we emphasize that the motivation and intervention of the coach could have been a limitation causing both groups to have worked indirectly under the same task conditions. dual and that is why both groups have improved at a motor but not cognitive level since apart from this handicap we found that the cognitive task was not challenging enough for the participants, in addition to having no similarity with what was measured in the Barcelona test.
Comment 12: Practical Implications: The discussion could explore the practical implications of the findings for designing training programs for older adults. For example, what specific components of the training program were most effective, and how can these be incorporated into real-world settings?
Response 12: We appreciate your comment since thanks to it we have contributed quality to the discussion by introducing a paragraph in which we point out what we should introduce in future training programs and what things should be improved (page 11 line 501-509).
Comments on the Quality of English Language
Comment 13: Overall, the document is well-written, but addressing these points would improve clarity, readability, and consistency, making it more accessible to a broader audience. Maintaining consistency in terminology would improve clarity.
Response 13: We appreciate your comment and we have tried to unify all the terms to give consistency to the language. It has also been reviewed by a translator following the comments that the reviewers have made to us.
Grammatical error
Comment 14: Example: There are a few grammatical errors and typos that should be corrected. For example, "an 8-week cognitive-motor training program on postural control and knee proprioception under single and dual-task in older adults" should be "an 8-week cognitive-motor training program on postural control and knee proprioception under single and dual tasks in older adults."
Response 14: We appreciate your comment and we add the modification to the text.

Round 2
Reviewer 2 Report
Comments and Suggestions for Authors
Many of the comments have still not been addressed or corrected.
Author Response
Dear Reviewer 2:
First of all, we would like to express our sincere gratitude for your thoughtful and detailed first review of our manuscript. Your input greatly contributed to improving the clarity and quality of our work. We carefully addressed each of your comments at the time and truly appreciate the time you have taken to review our responses and the revised manuscript.
However, we were concerned when you indicated that "many of the comments have not been addressed or corrected" without providing specific or concrete feedback on the areas that remain unsatisfactory. We want to ensure that we have adequately addressed all of your valuable concerns and that no points have been overlooked or misinterpreted. Below we describe the changes and clarifications we have made in response to your comments, highlighting key points for further consideration:
We have responded to all of the comments you made in the initial review with comprehensive changes or explanations. For example:
- Clarifications and expansions to the introduction and methodology sections (e.g., L69, L74, L76, L84) to include more context and references.
- Rationale for participant selection, details on exercise protocols, and references provided where necessary (e.g., L120, L150, L160).
- Inclusion of information on rest periods and clarifications on exercise and testing procedures (e.g., L177, L217, L241).
We have added and updated the bibliography, including references where requested (e.g., L150, L177, L248) and clarified the rationale behind key methodological choices.
The introduction was restructured and we refined the problem statement to address concerns about clarity and contextual relevance (e.g., L96-101, L107, L202-204).
We expanded the results section and included details about the statistical analysis (e.g., L283, L296, L304). Additionally, we clarified the findings and revised wording to avoid misunderstandings (e.g., L278, L330).
Expanded participant characteristics and study limitations were included to address any discrepancies in understanding (e.g., L278, L338-339).
We hope that these summaries help clarify that each point raised in your review has been thoroughly examined and addressed in our responses and reviews. However, if there are specific areas where you feel your comments were not adequately considered, we ask that you provide us with more detailed feedback. This will allow us to make fine-tuning adjustments and further refine our manuscript to your satisfaction.
We deeply value your expertise and are committed to incorporating any additional revisions that may be necessary. Thank you for your continued support and guidance during this review process.
With gratitude and respect,